# Improvement and Re-Evolution of Tetraploid Wheat for Global Environmental Challenge and Diversity Consumption Demand

**DOI:** 10.3390/ijms23042206

**Published:** 2022-02-17

**Authors:** Fan Yang, Jingjuan Zhang, Qier Liu, Hang Liu, Yonghong Zhou, Wuyun Yang, Wujun Ma

**Affiliations:** 1Australia-China Joint Centre for Wheat Improvement, College of Science, Health, Engineering and Education, Murdoch University, Perth, WA 6150, Australia; abtcyf@126.com (F.Y.); j.zhang@murdoch.edu.au (J.Z.); qier.liu@murdoch.edu.au (Q.L.); Hang.Liu@murdoch.edu.au (H.L.); 2Triticeae Research Institute, Sichuan Agricultural University, Chengdu 611130, China; zhouyh@sicau.edu.cn; 3Crop Research Institute, Sichuan Academy of Agricultural Sciences (SAAS), Chengdu 610066, China; 4College of Agronomy, Qingdao Agricultural University, Qingdao 266109, China

**Keywords:** wheat domestication, genetic diversity, durum wheat, improvement and re-evolution, introgressive hybridization

## Abstract

Allotetraploid durum wheat is the second most widely cultivated wheat, following hexaploid bread wheat, and is one of the major protein and calorie sources of the human diet. However, durum wheat is encountered with a severe grain yield bottleneck due to the erosion of genetic diversity stemming from long-term domestication and especially modern breeding programs. The improvement of yield and grain quality of durum wheat is crucial when confronted with the increasing global population, changing climate environments, and the non-ignorable increasing incidence of wheat-related disorders. This review summarized the domestication and evolution process and discussed the durum wheat re-evolution attempts performed by global researchers using diploid einkorn, tetraploid emmer wheat, hexaploid wheat (particularly the D-subgenome), etc. In addition, the re-evolution of durum wheat would be promoted by the genetic enrichment process, which could diversify allelic combinations through enhancing chromosome recombination (pentaploid hybridization or pairing of homologous chromosomes gene *Ph* mutant line induced homoeologous recombination) and environmental adaptability via alien introgressive genes (wide cross or distant hybridization followed by embryo rescue), and modifying target genes or traits by molecular approaches, such as CRISPR/Cas9 or RNA interference (RNAi). A brief discussion of the future perspectives for exploring germplasm for the modern improvement and re-evolution of durum wheat is included.

## 1. Introduction

Wheat is one of the main cereal crops in the world and a major source of carbohydrates and proteins in the human diet [1,2,3]. An adequate supply of wheat and its processed products (i.e., flour, bread, pasta, biscuit, bear) [4,5,6] can not only ensure food security and social stability but also enrich dietary diversity and avoid malnutrition [2,7,8]. The forecasted world population of 9.6 billion people in 2050 requires increasing wheat production by 60% in the next 30 years [9,10]. Moreover, the ever-changing climate brings great challenges to the sustainability of global wheat production [11,12,13,14,15,16]. Hence, the improvement of wheat on yield, quality, biotic resistance, abiotic tolerance, and diversity of the processed products has become a paramount task to cope with global climate change and to meet diverse consumption demands.

Wheat cultivars are usually divided into two groups, including tetraploid durum wheat (*Triticum turgidum* L. concv. *durum*, 2n = 4x = 28, AABB) and hexaploid bread wheat (*Triticum aestivum* L., 2n = 6x = 42, AABBDD) [17]. Allohexaploid wheat was derived from the spontaneous hybridization between tetraploid wheat and diploid *Aegilops tauschii* Cosson (2n = 2x = 14, DD) [18,19,20]. The greatly improved plasticity and adaptability of bread wheat conferred by the addition of the D genomes promotes it as one of the major cultivated crops in the world, accounting for approximately 95% of global wheat production [21,22]. Compared to bread wheat, durum wheat only accounts for the other 5%. Nevertheless, it is widely cultivated in many countries, including Turkey, Canada, Algeria, Italy, India, Australia, etc., with approximately 42.7 million tons of global production in 2020–2021 [23,24]. Compared to the other subspecies of tetraploid wheat, durum wheat is predominant in cultivation and food production due to its highly desirable phenotypes and diversified products for human consumption [24,25,26].

So far, many studies have reported the improvement of bread wheat performance through the introgression of the genetic variations of tetraploid wheat. Meanwhile, a few research works have reported the re-evolution of durum wheat through utilizing genes from other tetraploid subspecies or hexaploid wheat (particularly mediating the introgression of D genomes) [22,27,28]. This review summarizes the origin, domestication, evolution, and breeding strategies and achievements of tetraploid wheat, especially durum wheat. The updated information can shed some light on the future improvement of durum wheat to cope with global environmental challenges, meet the diversity consumption demands, and reduce wheat-related disorders in humans.

## 2. Origin, Domestication, and Evolution of Tetraploid Wheat

### 2.1. Origin Process of Tetraploid Wheat

Tetraploid wheat is constituted by two groups: Timopheevi group wheat (*Triticum timopheevi* Zhuk., 2n = 4x = 28, A^u^A^u^B^sp^B^sp^/A^u^A^u^GG) and Turgidum group wheat (*Triticum turgidum* L., 2n = 4x = 28, A^u^A^u^BB) (Table 1), according to the Biosystematics of Triticeae updated by Yen and Yang [17]. The origin of allotetraploid wheat can be traced back to presumably 0.3–0.5 million years before present (BP) in or near the oak-pistachio woodland belt, also called Near Eastern Fertile Crescent (Figure 1) [29,30,31].

Wild emmer wheat (*T. turgidum* var. *dicoccoides*, 2n = 4x = 28, A^u^A^u^BB) with brittle rachis (Br) and hulled seeds originated from the natural hybridization between two diploid ancestors, including wild einkorn wheat *Triticum urartu* (*Triticum monococcum* ssp. *urartu*, 2n = 2x = 14, A^u^A^u^) that provided the A genome [32,33], and a wild species of the *Aegilops* genus provided the B genome [34]. The donor of the A-genome for all tetraploid and hexaploid wheat was proposed to be another wild einkorn wheat *Triticum boeoticum* (*Triticum monococcum* ssp. *aegilopoides*, 2n = 2x = 14, A^b^A^b^), the ancestor of cultivated einkorn *Triticum monococcum* (2n = 2x = 14, A^m^A^m^), according to the early cytogenetic investigation, whereas it was proved to be the *T. urartu* later [31,32,33,35,36,37,38,39]. Although the origin of the B genome is still uncertain, immense geographical, morphological, cytological, genetic, biochemical, and molecular evidence has been accumulated, suggesting *Aegilops speltoides* (SS) as the contributor of the B genome [17,34,40,41,42,43,44]. However, recent research indicated that *Ae. speltoides* was not the direct progenitor of the B genome [45]. The comparison of genome sequences of five *Aegilops Sitopsis* species showed that *Ae. speltoides* and the B-subgenome diverged about 4.49 million years BP, which was much earlier than the speciation of tetraploid emmer wheat [45]. The long evolution history after tetraploid wheat generation may result in the high divergence of B genomes between hexaploid and diploid donors, making it difficult to deduce the precise donor/s [46,47].

*T. araraticum* (*T. timopheevi* var. *araraticum*, 2n = 4x = 28, A^u^A^u^B^sp^B^sp^) was generated in another independent polyploidization event between *T. urartu* and other subspecies of *Ae. speltoides* (B^sp^B^sp^/GG), representing the wild forms of timopheevi wheat [41,48,49,50]. It is indistinguishable from *T. dicoccoides* in morphology but different in genomic constitution, therefore showing irregular meiosis and high infertility when crossed with *T. turgidum* group wheat or hexaploid wheat [17,29].

Wild emmer is native to the Fertile Crescent and is about 0.36 million years old [51]. It was at 19 thousand years BP that hunter–gatherers started to collect and use wild emmer wheat based on the archaeological evidence found at Ohalo II located on the southwestern shore of the Sea of Galilee, Israel [52,53]. The first spikelet of wild emmer wheat was isolated by T. Kotschy from specimens of wild barley (*Hordeum spontaneum* K.) in 1855 and recognized as wild wheat by Kornicke until 1873 [20,54]. In 1906, the wild emmer wheat was rediscovered in nature by Aaron Aaronsohn near Rosh Pinna, eastern Galilee [55]. According to the literature, this species is still distributed in present-day in the Jordan valley, Southeastern Turkey, Eastern Iraq, and Western Iran [29,54]. It is usually growing in various geological conditions, such as basalt areas, hard limestone bedrocks, Terra Rossa soils, and grass and woodland of the hill country, with altitudes ranging from about 150 m below sea level to 1800 m above sea level [29,54]. The complicated grow habitats might promote the genetic differentiation and evolution of wild emmer wheat.

### 2.2. Domestication and Evolution Process

The domestication of wild emmer wheat started about 10 thousand years BP (Figure 1), which symbolized the first and significant evolution intervention by humans and raised the Neolithic revolution [20,56]. Consequently, two cultivated tetraploid wheat, *T. turgidum* concv. *dicoccon* (2n = 4x = 28, A^u^A^u^BB) and *T. timopheevi* concv. *timopheevi* (2n = 4x = 28, A^u^A^u^B^sp^B^sp^), were domesticated from *T. dicoccoides* and *T. araraticum*, respectively [29,57,58,59,60,61,62]. Since the birth of domesticated emmer wheat in the Fertile Crescent, *T. dicoccon* has been widely spread from farming villages in Pre-Pottery Neolithic B (PPNB) to the Mediterranean region and has become one of the most prominent crops for almost 6 thousand years [35,63]. Around 9 thousand years BP, allohexaploid Spelt wheat (*Triticum spelt*, 2n = 6x = 42, A^u^A^u^BBDD) was produced through the natural hybridization between *T. dicoccon* and *Ae. tauschii* [18,19,26]. In addition, the domesticated *T. timopheevii* naturally hybridized with *T. boeoticum* gave rise to the hexaploid Zhukoyskyi wheat (*Triticum zhukovskyi*, 2n = 6x = 42, A^u^A^u^B^sp^B^sp^A^b^A^b^) in Transcaucasia [26,64]. About 8.5 thousand years BP, the currently most cultivated durum wheat (*T. turgidum* concv. *durum*) with a tough rachis and free-threshing seeds was produced from natural mutation of *T. dicoccon* in the Eastern Mediterranean region (Figure 1) [52,63]. The subsequential polyploidization between domesticated emmer *T. dicoccon* (or cultivated *T. durum*) and *Ae. tauschii* Coss. gave rise to bread wheat [20,26]. Additionally, some research indicated that bread wheat was produced by the mutation of *Triticum spelt* [20,26].

According to the results of morphological and molecular evidence, two distinct populations of *T. dicoccoides* have been identified in Southern and Northern Levantine Corridor sites, resulting in dissident opinions on the geography of domestication [52,63,65,66]. The amplified fragment length polymorphism (AFLP) marker analysis was initially used to compare the wild and domesticated emmer wheat and found that most of the wild emmer lines collected from the Karacadag mountain of southeastern Turkey were more related to domesticated emmer and durum wheat, laying the foundation of monophyletic origin opinion [67]. The re-analysis of the AFLP data through principal coordinate analysis (PCA) supported the previous result of Özkan [68]. Further research proposed an independent domestication event of wild emmer wheat that occurred in Karacadag mountain and somewhere else in southeastern Turkey, inferring from the result of chloroplast DNA fingerprinting [65]. Subsequently, larger populations were used to reconsider the domestication geography of tetraploid wheat [69]. The authors suggested that the central-eastern race was the progenitor of domesticated emmer wheat since it played a key role during domestication. They also declared disagreement evidence on the domestication origin site based on the data of chloroplast DNA (Kartal-Karadag mountain) and AFLP analysis (Karacadag range) [69]. In 2007, more tetraploid accessions were investigated by Luo et al. [63], who agreed with the monophyletic domestication in Northern Levant followed by subsequent hybridization and introgression from wild to domesticated emmer in the southern Levant. Another possibility is that *T. dicoccoides* was independently domesticated in the Northern and Southern Levant [63]. Archaeobotanical findings are consistent with the polycentric domestication model of wild emmer wheat across the Levant [52]. Following this research, Özkan et al. [54] suggested that there was a pre-adaption wild emmer race spreading to several locations in the Fertile Crescent before being domesticated. So far, the theory of multiple site-independent domestications of *T. dicoccoides* across the Levant might be more realistic [48]. Accordingly, multiple genes and traits were transferred and involved in selection and evolution through numerous hybridizations and mutations. Consequently, polymorphic populations, rather than single genotypes, evolved from the domesticated emmer wheat, such as Rivet wheat (*T. turgidum* concv. *turgidum*), Polish wheat (*T. turgidum* concv. *polonicum*), and Persian wheat (*T. turgidum* concv. *carthlicum*) [17,58].

### 2.3. Variations of Major Traits during Domestication and Evolution

Crop domestication aims to meet human needs and environmental conditions. Wild wheat, barley, and rye were preferred over the other cereals for ancient people, attributed to the large spikes with large and heavy grains [29,52]. Furthermore, the domestication of wild emmer was much better than barley since the former has more and larger grains [14]. The purposeful selection of domestication accompanied by the stress of specific agroecological environments, natural hybridization, and mutation has led to the ‘domestication syndrome,’ including genomic, morphologic, and phenotypic variations. The evolution from brittle rachis, tough glumes, and hulled seeds to non-shattering, soft glumes, and free-threshing might be the primary domestication targets of wild emmer wheat, as the processes facilitated harvesting by the primitive farmer [31,48].

#### 2.3.1. Changes from Brittle to Non-Brittle Rachis

The brittle rachis (Br) promotes wild tetraploid wheat to disperse seeds freely at maturity through developing abscission of fracture zone at the joint of articulation of the spikelet and rachis, which results in high yield reduction [70]. However, the tough rachis of domesticated forms of emmer wheat suppressed seed dispersal and self-planting and made grain harvesting feasible. Hence, the target transformation of spikes from Br to non-Br was constantly conducted by early farmers for more than one thousand years, symbolizing the first trait of domestication in wheat [56,71,72]. This qualitative trait modification is critical for the origin of agriculture and sedentary societies [20].

The dominant *Br* genes have been mapped on the short arms of the chromosomes of homologous group 3 in both tetraploid and hexaploid wheat [73,74,75,76] and their key roles have been identified [77,78,79]. At first, a single dominant gene, *Br1* (*Br-A1*), was identified that controlled rachis fragility in feral or semi-wild hexaploid wheat from Tibet [80]. This gene was mapped on the short arm of the 3D chromosome later [74]. Two other dominant genes, *Br2* (*Br-A2*) and *Br3* (*Br-A3*), controlled the Br phenotypes in wild emmer wheat and were localized on the short arms of 3A and 3B chromosomes, respectively [59,81]. Three types of rachis fragility, semi-wild wheat type, spelta-type, and tough rachis type, were controlled by the interaction of these three genes [20]. The haplotype analysis of *Br1-A* and *Br1-B* genes confirmed the common origin of the cultivated Turgidum group wheat and indicated a separate domestication event of *T. timopheevii* [61]. The recessive alleles from mutant *Br* genes conferred non-shattering spikes to domesticated emmer, facilitating harvest [20,26].

#### 2.3.2. Variations from Non-Free Threshing to Free Threshing

The grains of both wild and domesticated tetraploid wheat are wrapped with tough glumes (Tg), consequently making them hard to thresh [82]. On the contrary, all cultivated tetraploid wheat is characterized by soft glumes and free-threshing (non-hulled seeds) [31,83]. The evolution of the free-threshing trait in durum wheat was mainly attributed to the mutation of the semi-dominant *Tg* gene on the short arms of 2A and 2B and another critical domestication dominant gene, *Q* (primitive allele as *q*), located on the long arm of the 5A chromosome [83,84,85,86,87,88,89,90,91,92,93]. Compared with the *Q* gene, *Tg* showed a more pronounced effect on the threshability of seeds [85]. All cultivated tetraploid wheat that display the free-threshing phenotype should carry *QQtgtg* combinations. Therefore, the *QQTgTg* genotype might be concealed in the domestication process of emmer wheat [26].

The *Q* gene originated from a single amino acid mutation from the wild type of the *q* allele and encoded a transcription factor belonging to the APETALA2 (AP2) family [89,94]. The variations between *Q* and *q* alleles took place in the conserved coding region, from A to G at position 985 and from C to T at position 1254. The *Q* gene enhanced protein dimerization activity relative to *q* and reduced the binding and degradation of *microRNA172*, resulting in higher transcription levels, which lead to a compact spike with a free-threshing phenotype [95,96]. The fragile rachides of wild wheat and the tough rachis of the semi-dominant mutant indicate the other role of the *Q* gene in controlling the trait of brittle rachis [97]. In addition, dosage effects of *Q* and *q* alleles were observed on the spike phenotype [98]. The loss-of-function mutant of *Q* and *q* genes led to a speltoid-like spike and non-free-threshing phenotype [99]. Considering the pleiotropic effects of the *Q* gene on rachis fragility, glume tenacity, threshability, spike architecture, plant height, and flowering time, it could be considered a gain-of-function mutation and recognized as the most important domestication gene [83,94,96,99,100,101].

#### 2.3.3. Other Qualitative Domestication Traits Involved

During the domestication process, many other traits were co-selected, including plant height, tiller number, flowering/heading time, phenotypes of the spike, spikelet, seed, etc. [31,48,71,83,96,102,103,104,105,106,107]. For example, spring wheat lacked vernalization and specific photoperiod requirements were domesticated from winter wheat for good adaptation to the prevailing environmental conditions in the Fertile Crescent region [20]. The evolution of this adaptation model plays a key role in the post-domestication for the adaptation of temperate cereals [108]. In tetraploid wheat, the QTLs related to flowering/heading time were mapped on 2A, 4B, 5A, and 6B [71]. Among them, the wild allele for the QTL on 5A is associated with late-anthesis of wild emmer, whereas the other three QTL are responsible for early flowering [71]. The QTL mapped on 2A is present in a collinear position with the photoperiod response (*Ppd*) genes (2AS, 2BS, 2DS) [48]. Plant height affects lodging and consequently grain yield and quality traits [109]. Modern dwarf wheat was attributed to the reduced height gene *Rht1* (*Rht-B1b*) (located on 4B chromosomes), which can substantially reduce the height of wheat, prevent plant lodging, and increase the harvest index [109,110,111]. Many dwarfing genes have been mapped on the AB genomes of wheat and classified into the phytohormone gibberellin (GA)-insensitive (i.e., *Rht1*, *Rht3* (*Rht-B1c*), and *Rht11* (*RhtB1e*)) and GA-sensitive (i.e., *Rht9*, *Rht18*, and *Rht24*) groups based upon the plant response to bioactive GA [112]. Recently, a recessive semi-dwarfing gene *Rht-dp* was fine mapped in a Polish wheat line, which was subsequently identified as the same gene of Green Revolution contributor *Rht1* [112,113]. These evolved qualitative and quantitative traits under the driving force of domestication had a substantial impact on the early cultivation of cereal crops and the rise of modern agriculture [99].

### 2.4. Importance and Breeding Challenge of Durum Wheat

After thousands of years of empirical selection and breeding, modern cultivated tetraploid wheat and hexaploid wheat were formed and widely cultivated, accompanying morphological, phenotypical, and physiological changes [31,48,71]. The domestication and evolution led to modern cultivated tetraploid and hexaploid wheat superseding einkorn, emmer, and spelt wheat, which are treated as relic crops with minor economic importance [1,21]. Einkorn was used at the beginning of agriculture in Europe, whereas its cultivation started to decline in the Bronze Age, probably due to the emergence of free-threshing wheat [60,114]. Although the free-threshing form of einkorn was discovered as well, its cultivation area was limited since the soft glume was associated with the reduction of ear length and grain yield [60]. The cultivation of the tetraploid *T. timopheevii* wheat was limited as well because of its hard threshing trait and infertility of hybrids when crossed with *T. turdidum* or *T. aestivum* [49,61]. Among the different tetraploid wheat, durum wheat is currently the most widely cultivated subspecies, particularly in the Mediterranean region, which accounts for more than half of the worldwide growing region of durum wheat [25]. There are probably a couple of reasons for the durum wheat adaptation. First, along with the einkorn wheat declination, durum wheat had almost completely substituted the wild and domesticated emmer wheat based on its ideal phenotypes (tough rachis, soft glumes, free-threshing ability, large seeds, and short dormancy, etc.) at the beginning of the 20th century [115,116]. Exceptionally, *T. dicoccum* is still the main food resource for Romans in Italy [115]. Second, durum wheat can be used for diverse food products in terms of its specific characterizations [117,118]. Pasta is the most common end product of durum wheat and supplied 15.8 million tons in the world in 2019 [119]. In addition, the evolution of durum wheat has gone through a long history from wild emmer, domesticated emmer, cultivated durum landraces, to modern durum cultivars, accompanying allopolyploidy from diploid to tetraploid [24,120], which makes durum wheat an excellent research model for the evolution of allopolyploid speciation, adaptation and domestication in plants [121]. Moreover, durum wheat is cross-compatible and inter-fertile with bread wheat since they share the same AB genomes [122].

However, undesirable domestication syndrome of cultivated tetraploid wheat exists. Accompanied with domestication and especially breeding programs of modern agriculture with limited primal parents, the genetic diversity of current cultivars dramatically declined, which in turn led modern cultivars to be susceptible and vulnerable to biotic or abiotic stress [120,123]. Haudry et al. [124] revealed that during domestication, the genetic diversity in the cultivated forms was reduced by 84% in durum wheat and 69% in bread wheat. Genomic comparative analysis between Svevo (modern durum wheat cultivar) and Zavitan (wild emmer wheat) revealed that regions exhibiting strong signatures of genetic divergence (associated with domestication and breeding) were widespread in the genome, with several major diversity losses in the pericentromeric regions [120]. They also identified a high-cadmium accumulation allele widespread among durum cultivars but undetected in wild emmer accessions, which was caused by a non-functional variant of a metal transporter *TdHMA3-B1* encoding gene [120]. Therefore, it is a paramount task to reinforce the genetic diversity of durum wheat and improve its yield, grain quality, biotic resistance, and abiotic tolerance to cope with the rapidly growing population and the challenges of global warming [9,10,11,12,13,14,15,16,125].

## 3. Improvement and Re-Evolution of Durum Wheat

### 3.1. Strategies and Approaches

Since wild emmer wheat contains numerous favorable genes and shows complete genomic compatibility with durum wheat, it can be directly used to enrich the genetic diversity of durum wheat [126,127]. *Aegilops* species also contain numerous useful genes [128]. Compared to hexaploid wheat with the addition of the D genomes, durum wheat has limited genetic variations and allelic combinations, resulting in narrow adaptability to different photoperiods, vernalization, and soil conditions. These characterizations inserted a negative impact on yield potential, end-use quality, and capacities to make various food products [21,129,130]. Considering the great effects of the D genome of *Ae. tauschii*, researchers also proposed to introduce the D genome to re-evolve durum wheat through hybridizing with hexaploid wheat that can readily produce hybrids [21,22,28,122,129]. In diploid wheat, the useful traits from the sources of einkorn wheat other than the D genome can also be integrated into durum wheat smoothly [131,132].

To date, a wide range of approaches have been used for the improvement of durum wheat, as summarized in Table 2. Hybridization directly with diploid einkorn, wild emmer, and hexaploid wheat has been the most traditional breeding method [20,122,133]. Similarly, inactivation of the pairing of homologous chromosomes gene *Ph* located on the 5B chromosome can induce pairing of homoeologous chromosomes for mediating genetic introgression in wheat-alien hybrid combinations. The most common mutants include durum wheat *ph1c* and bread wheat *ph1b* [134,135,136]. Moreover, Joppa developed a set of disomic substitution lines, in which the chromosome pairs of tetraploid wheat Langdon were replaced by the homoeologous D-chromosome pair from Chinese Spring (CS, a hexaploid wheat landrace) [137,138]. These substitution lines have been widely used to integrate the D-genome into durum wheat cultivars. On the other hand, with the development of modern biotechnology, speedy and effective breeding of durum wheat can be achieved through molecular marker-assisted selection (MAS), genome sequencing-assisted breeding, and target gene modification, such as RNA interference (RNAi) and CRISPR-Cas9, leading to many durum wheat lines with improved agronomic traits, protein quality, biotic resistance, abiotic tolerance, and reduced or eliminated triggering factors of wheat plant disorders [139,140,141,142,143,144].

### 3.2. Major Achievements

#### 3.2.1. Crop Yield Potential

Crop yield is the main goal of durum wheat breeding associated with multiple component traits, such as plant height, tiller number, spike number, spike length, spikelet number per spike, kernel number per spike, and thousand-grain weight. Semidwarf cultivars of durum wheat were generated by introducing the *Rht1* (*Rht-B1b*) gene from Norin 10, leading to a significantly increased harvest index [145]. Compared to durum wheat, bread wheat can utilize more dwarf genes, such as the homoeologous gene *Rht2* (*Rht-D1b*) on the 4D chromosome [109]. Recent research revealed that *Rht1* could reduce plant height by generating an N-terminal truncated DELLA protein, the key repressors of the GA signaling pathway and plant growth, through tissue-specific translational reinitiating, which has no effect on the dormancy of seeds [189]. In durum, two dwarfing genes, *Rht14* and *Rht15*, have been introgressed into cultivar Langdon through crossing with dwarf lines [146,147]. However, the introgressed lines showed a decrease in yield component traits. By crossing durum wheat with hexaploid wheat, some tetraploid lines were developed with largely increased spike length, spikelet per spike, grain number per spike, and thousand-grain weight through introgressing the genome of hexaploid wheat using inter-ploidy hybridization [148]. Recently, pentaploid hybridization has also been used to cross durum wheat with its nascent synthetic hexaploid wheat and developed a durum wheat—*Ae. tauschii* Coss. 4D (4B) disomic substitution line YL-443 [149]. Compared to the tetraploid parent, the developed line showed a larger spike with an increased number of spikelets and florets per spike by 36.3 and 75.9%, respectively [149]. In addition, the kernel size of durum wheat could be significantly increased by knocking out the grain weight gene *GW2* through RNAi technology, leading to increased grain starch content by 10–40% [150].

#### 3.2.2. Grain Quality

The grain quality of durum wheat is usually related to kernel hardness, grain protein content, dough tenacity and extensibility, gluten strength, yellow pigment concentration, etc. [117,190]. The kernel texture of hexaploid wheat is controlled by the *Hardness* locus located on the short arm of the 5D chromosome, which contains two puroindoline genes, *Pina-D1* and *Pinb-D1*, and the grain softness protein gene *Gsp-D1* [191]. In diploid wheat, these three genes of the *Hardness* locus are harbored on 5A or 5B chromosomes, whereas the *Pin* genes were deleted from these two chromosomes in durum wheat, resulting in the loss of the softness-conferring PIN proteins and consequently a generally hard texture [192]. The soft kernel character of hexaploid wheat contributed to the reintroduction of the *Pin* genes from *Ae. tauschii* Coss. [193]. Nevertheless, soft kernel texture of durum wheat with improved baking quality was observed through introducing the *Hardness* locus from the 5D chromosome of bread wheat, of which the hybridization was mediated by *ph1b* or *ph1c* lines [28,151,152]. By crossing the *Pina*-expressing transgenic line with high molecular weight (HMW) glutenin subunit 1A × 1-expressing transgenic line of durum wheat, a few 1A × 1 and *Pina* co-expressing lines were generated, showing a significantly improved breadmaking quality and unaffected pasta-making potential [153]. Compared to the transgenic lines, which only express 1A × 1 or *Pina*, the 1A × 1 compensated for the detrimental effect of the *Pina* gene (reduce water absorption and damage starch) in co-expressing lines [153]. High protein content and pure durum semolina are strictly required, especially in Italy, France, and Greece, for manufacturing pasta or some bakery products [190]. The HMW glutenin encoding locus *Glu-D1* was transferred from 1D of common wheat into 1A of the recipient durum wheat and greatly improved bread-making quality of the latter [28,154,155,156,157,158,159]. It was concluded that the Dx2 + Dy12 was superior to another *Glu-D1* allelic variant Dx5 + Dy10 on bread-making quality in durum wheat, as the latter produced excessively strong and inelastic doughs [194]. Moreover, the grain protein content gene *Gpc-B1*, a NAC (NAM, ATAF1/2, CUC2) transcription factor for high grain protein, zinc, and iron contents, was successfully introgressed into two high-yielding but lower protein Canadian durum wheat lines [126,160].

Yellow pigment concentration represents the carotenoid accumulation in the kernels of durum wheat and has been regarded as a source of important nutrients or antioxidant compounds [195]. A DNA marker PSY-1SSR for yellow pigment concentration QTL, *Qyp.macs-7A*, was developed and used for marker assistant selection in durum wheat breeding [196]. Multiple mutant combinations of the *β*-carotene synthetic key inhibitor genes, *lycopene ε-cyclase* (*LCYe*) and *β-carotene hydroxylase 2* (*HYD2*), were isolated by Yu et al. [161] through Targeting Induced Local Lesions in Genomes (TILLING), which significantly increased *β*-carotene in the endosperm for biofortification of provitamin A in durum wheat.

In addition, pre-harvest sprouting could significantly degrade the end-use quality of durum wheat and cause a huge reduction in yield, occurring especially in cool and moist conditions after maturity [162]. A major QTL, *Qphs.sicau-3B.1*, was introgressed from the long arm of the 3B chromosome in hexaploid wheat *T. spelta* into durum wheat, conferring its high resistance to pre-harvest sprouting [162]. Sequence characterized amplified region (SCAR) markers were also developed for tracking and breeding durum wheat cultivars with resistance to pre-harvest sprouting, higher yield, and good end-use quality [162].

#### 3.2.3. Biotic Resistance

Wheat grows in diverse geographical regions and environments under different production systems, making it exposed to various biological pathogens that might cause a huge loss of yield or decrease in grain quality [197]. Common diseases include leaf rust (LR), stripe rust (YR), stem rust (SR), powdery mildew (PM), fusarium head blight (FHB), etc. [197]. Through crossing with resistant durum wheat or hexaploid wheat, the *Fhb1*, *Pm13*, and *Lr19* (tightly linked with yellow pigment in endosperm) genes and some disease-resistance QTLs were successfully transferred into durum wheat [156,163,164,171,172,173,174]. It was reported that some durum wheat lines carried multiple improved traits, such as resistance to leaf rust (*Lr14*) and tan spot (races 4 and 6), higher grain yield, and higher grain protein content, through crossing with *T. araraticum*, *T. dicoccoides*, and *Ae. speltoides* [165]. By introducing resistance genes *Lr34/Yr18/Sr57/Pm38/Ltn1* from hexaploid wheat, durum wheat was conferred with a robust seedling resistance to leaf rust, stripe rust, and powdery mildew diseases [166]. The resistant genes of stem rust (*Sr22*) and stripe rust (none specify) introgressed from the A genome of diploids enhanced the disease resistance of durum wheat [167,168].

Synthetic wheat is another way to introduce biotic resistance genes to durum wheat. The synthesized resistant AABBAA amphiploid of *T. durum* and *T. boeoticum*, *T. monococcum*, or *T. urartu* are useful genetic resources for stripe rust, leaf rust, and powdery mildew resistance, which could be transferred into cultivated durum wheat [168,169,170]. Previous studies indicated that the stripe rust resistance gene *Yr28* of durum wheat-*Ae. tauschii* 4D (4B) disomic substitution line could be fully expressed in the tetraploid background, whereas partially expressed in the hexaploid background, which can be used for improving stripe rust resistance in durum wheat breeding [149]. These results demonstrate the successful synthetic methodology in durum wheat breeding programs.

In addition, advanced durum wheat lines with promising FHB resistance were screened out through treatment with 5-azacytidine (DNA methylation inhibitor), which might be contributed by a significant number of differentially expressed genes related to the biosynthesis of secondary metabolites, MAPK signaling, photosynthesis, etc. [175]. Hessian fly is a severe pest of winter wheat, resulting in a reduction of grain and forage production through stunting and killing vegetative tillers, preventing spike developing, and reducing grain filling [198]. Increased resistance to Hessian fly together with superior agro-phenological traits have been obtained in recombinant inbred durum wheat lines [176].

#### 3.2.4. Abiotic Tolerance

Environmental stress, such as drought, high or low temperature, and soil salinity, pose a great threat to the production of durum wheat. Enhancing the environmental adaptability of durum wheat has great potential for increasing its yield and quality. The thermotolerance of durum wheat could be improved by inducing the expression of heat shock proteins (HSPs) [177,178]. By introgression of alien genes from wild emmer, durum wheat displayed a significant increase in the root-to-shoot ratio for enhancing water stress tolerance [179]. Overexpression of *TdPIP2;1*, a wheat aquaporin gene, could significantly enhance drought and salt tolerance in transgenic lines of durum wheat by reducing excessive water evaporation from leaves in response to water deficit [180]. The ascorbic acid (AsA) treatment could also be useful in enhancing the salt tolerance of durum wheat, whereas it might not be suitable for field use [181]. In bread wheat, the major Al^3+^ tolerance *Kna1* locus and *TaALMT1* gene are mapped on the 4D chromosome [199,200]. The *ph1*-mediated pentaploid hybridization and intraspecific crosses have been used to transfer these genes or loci from bread wheat and Langdon-CS 4D (4B) substitution line, respectively, into durum wheat for developing advanced tolerant lines [182,183,184]. Other than the D genome resistant sources, a wide range of crosses with subspecies of the turgidum group of wheat (emmer, durum, carthlicum, turgidum, turanicum, polonicum) also improved durum wheat’s salt tolerance [185,186].

#### 3.2.5. Wheat-Related Disorders in Humans

During the last decades, an increasing incidence of wheat-related disorders has been reported in human beings worldwide, such as celiac disease (CD), wheat allergy (WA), wheat-dependent exercise-induced anaphylaxis (WDEIA), and non-celiac wheat sensitivity (NCWS) [201,202,203]. Currently, there are 28 allergens identified in wheat that can cause severe immune diseases [204]. CD is caused by the ingestion of gluten proteins from wheat, leading to an autoimmune disorder [205]. In bread wheat, the downregulation of gliadins (the main toxic component of gluten) has been successfully achieved through RNAi technology [206,207]. In durum wheat, the 33-mer, the main immunodominant peptide of four highly stimulatory peptides encoding by the α-gliadin genes, was precisely modified through the CRISPR/Cas9 technology [187]. Compared to the wild type, the produced mutant lines showed a highest 69% reduction of gliadin in the seeds, displaying a reduced immunoreactivity [187]. On the other hand, the α-amylase/trypsin inhibitors (ATI) are the major triggering factors responsible for the onset of bakers’ asthma (develops after allergen inhalation) [208,209]. An ATI mutant durum wheat obtained through CRISPR/Cas9 multiplex editing knocked out two major related subunits, WTAI-CM3 and WTAI-CM16, showed reduced allergen proteins [188]. Compared to traditional breeding strategies, novel technologies focused on manipulating molecule levels tend to be more time-saving and effective, especially in modifying traits controlled by multiple genes and purging deleterious alleles.

## 4. Future Perspectives

Considering the substantial role that durum wheat has been playing in the global food supply, its yield and product diversity are related to a sustainable global development, especially upon facing severe climate change and rapidly growing populations. However, the long history of domestication and modern breeding programs have caused serious genetic erosion in durum wheat, leading to a bottleneck in yield improvement and vulnerability to biotic and abiotic stress. To broaden the genetic diversity of durum wheat, massive germplasm resources and approaches have been employed in the attempts of re-evolution of durum wheat as discussed above. Future improvement efforts should be focused on the following aspects.

First, durum wheat is a polyploid species that has experienced a long evolution. The artificial and natural selection caused huge variations between durum wheat and its subgenome donors. Species including *T. urartu*, *T. boeoticum*, *T. monococcum*, and *Ae. speltoides* are essential resources for the improvement of quality traits, resistance against several wheat diseases, and environmental adaptability. They could be used to transfer alien genes into durum wheat through inter-ploidy hybridization. Moreover, desirable genes or traits can also be introduced into durum wheat via ‘bridge-crossing’ using synthesized amphiploids, for instances AASS, AABBA^m^A^m^, and AABBSS [168,170,210,211,212,213].

Secondly, the tetraploid relatives of durum wheat and hexaploid wheat are critical germplasm for durum wheat breeding. For example, wild emmer wheat possesses many important beneficial traits, such as resistance to stripe rust, stem rust, and powdery mildew, and high tillering capacity, grain protein content, photosynthetic yield, and salt and drought tolerance [126]. The other cultivated turgidum wheat, i.e., Rivet wheat, Polish wheat, and Persian wheat, also contain useful genes [185]. The elite genes or traits carried by the ABD genomes of hexaploid wheat can be easily introgressed into durum wheat through crossing. In addition, the AAGG group wheat should also be explored for genetic enrichment of durum wheat [214].

Third, the availability of the whole genome of allohexaploid wheat CS and several allotetraploid wheat has eased the exploration of new genes in durum breeding. So far, numerous genes still have untapped functions. Conventional breeding should be combined with modern molecular biological technologies, such as CRISPR/Cas9 and RNAi. Through the introduction of the *TaWOX5* gene, the transformation efficiency of tetraploid Polish wheat has been dramatically enhanced, which makes gene-editing an ideal approach for durum wheat improvement [215]. Multiple desirable alleles can be pyramided to meet different breeding requirements [216]. On the other hand, deleterious genes or alleles need to be silenced or knocked out, such as the brittle rachis and tough glume carried by wild emmer wheat. The desirable candidate genes can be transferred into durum wheat without linkage drag of deleterious genes.

## Figures and Tables

**Figure 1 ijms-23-02206-f001:**
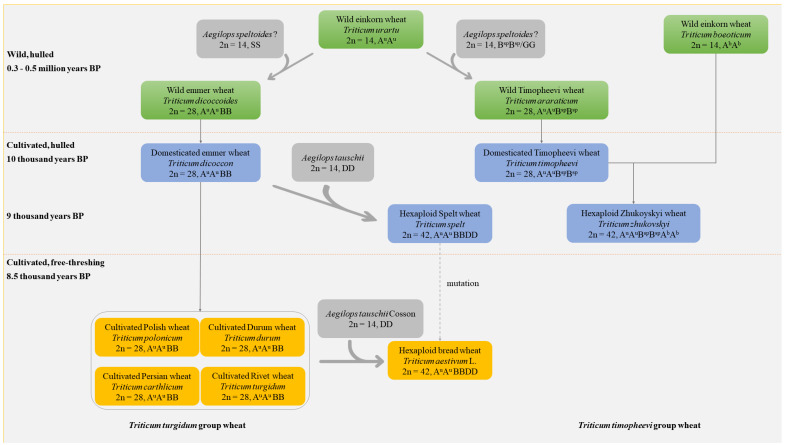
Evolution of *Triticum turgidum* and *Triticum timophevvi* group wheat. BP, before present. The dashed gray arrow denotes the possible but not confirmed pathway for the origin of bread wheat (*T. aestivum* L.).

**Table 1 ijms-23-02206-t001:** Botanical category of tetraploid wheat.

Ploidy	Categories	Botanical Name	Synonyms	Phenotype
Tetraploid 2n = 28	*T. timopheevi* Zhuk. A^u^A^u^B^sp^B^sp^/ A^u^A^u^GG	var. *araraticum*	*T. araraticum* Jakubz.	wild, brittle rachis, hulled
		concv. *timopheevi*	*T. timopheevi* Zhuk.	domesticated, hulled
	*T. turgidum* L. A^u^A^u^BB	var. *dicoccoides*	*T. dicoccoides* Koern.	wild, hulled, brittle rachis
		concv. *dicoccon/dicoccum*	*T. dicoccon* Schrank.	domesticated, hulled, semi-brittle rachis
		concv. *durum*	*T. durum* Desf.	cultivated, free-threshing, tough rachis
		concv. *turgidum*	*T. turgidum* L.	cultivated, free-threshing, tough rachis
		concv. *polonicum*	*T. polonicum* L.	cultivated, free-threshing, tough rachis
		concv. *carthlicum*	*T. carthlicum* Nevski.	cultivated, free-threshing, tough rachis

concv., cultivar-group; var., variety.

**Table 2 ijms-23-02206-t002:** Summary of the improvement of durum wheat through diversity germplasms and approaches.

Trait		Genes/QTLs	Chromosome	Approaches	Origin	Variations	References
Yield component	plant height	*Rht1*, *Rht14,* *Rht15*	4B, 4A, 6A	pentaploid hybridization, homologous hybridization	hexaploid wheat Norin10, durum wheat Castelporziano or Durox	reduced height; increased harvest index or pleiotropic effect	[145,146,147]
	multiple traits	/	/	pentaploid hybridization	hexaploid wheat CSCR6	pleiotropic effect	[148]
		/	4D	pentaploid hybridization	*Ae. tauschii* accession AT23	big spike, significantly increased number of spikelets and florets per spike; enhanced YR resistance	[149]
	grain size	*GW2-A1,* *GW2-B1*	6AS, 6BS	RNA interference	durum wheat Svevo	increased kernel size	[150]
Grain quality	soft kernel	*Pina,* *Pinb*	5D	*ph1c*-mediated homoeologous recombination	Langdon 5D(5B) substitution line, durum wheat *ph1c* line Cappelli M	soft grain, vitreous kernels, high GPC, and good gluten quality	[151]
				*ph1b*-mediated homoeologous recombination, homologous recombination	hexaploid wheat CS, Langdon-CS 5D(5B) substitution line	soft grain	[152]
		*Pina*, HMW glutenin subunit 1A × 1	1D, 1A	gene editing	durum wheat	soft kernel and better breadmaking quality	[153]
	flour properties	*Glu-D1*	1D	*ph1*-mediated homoeologous recombination	multiple germplasms	improved bread-making quality	[28,154,155,156,157,158,159]
	grain protein content	*Gpc-B1*	6B	homologous recombination	Langdon-*T. dicoccoides* (6B) substitution line	increased protein level	[160]
	yellow pigment concentration	*LCYe,* *HYD2*		Targeting Induced Local Lesions in Genomes (TILLING)	durum wheat	significantly increased *β*-carotene accumulation	[161]
	pre-harvest sprouting	*Qphs.sicau-3B.1*	3B	pentaploid hybridization	hexaploid wheat *T. spelta* CSCR6	high resistance to PHS	[162]
Biotic resistance	leaf rust, powdery mildew, tan spot	*Pm13*, *Lr19,* YPC	3B, 7A	*ph1* mediated homoeologous recombination	*Ae. Longissima,* *Agropyron*	improved resistance to PM, LR, and YPC	[156,163,164]
		multiple	/	inter/intra-specific hybridization	*T. araraticum*, *T. dicoccoides*, *Ae. speltoides*	enhanced resistance to LR and tan spot; increased grain yield and GPC	[165]
		*Lr34/Yr18/Sr57/Pm38/Ltn1*	/	gene editing	bread wheat	robust seedling resistance to LR, YR, and PM	[166]
	stem rust	*Sr22*	7A	interspecific hybridization	*T. monococcum* L. cv. RL 5244	differential resistance depends on ploidy level	[167]
	stripe rust	/	/	interspecific hybridization	A-genome diploids	enhanced resistance to YR, LR, and PM	[168,169,170]
		*Yr28*	4D	pentaploid hybridization	*Ae. tauschii* accession AT23	enhanced resistance to YR	[149]
	fusarium head blight	*Fhb1*	3BS, 4AL, 4BS, 5AL, 6AS	homologous recombination	hexaploid wheat Sumai-3	improved resistance to FHB	[171]
		*Qfhb.ndwp-5A, Qfhb.ndwp-7A*	5A,7A	homologous recombination	hexaploid wheat PI 277012		[172]
		/	2AS, 2BS, 3AL, 4BL	pentaploid hybridization	hexaploid wheat Sumai-3		[173]
		/	/	pentaploid hybridization	hexaploid wheat Sumai-3		[174]
		/	/	mutation by treating with DNA methylation inhibitor (5-methyl-azacytidine)	/		[175]
	Hessian fly	/	/	/	durum wheat resistant lines	enhanced resistance to Hessian fly, superior agro-phenological traits	[176]
Abiotic resistance	thermotolerance	heat shock proteins	/	gene modify (overexpression)	/	improved thermotolerance	[177,178]
	water	/	/	intraspecific hybridization	wild emmer	enhanced adaptation to water stress	[179]
	drought, salt	*TdPIP2;1*	/	gene modify (overexpression)	durum wheat	enhanced drought and salt tolerance	[180]
	salt	/	/	induced by ascorbic acid	/	enhanced salt tolerance	[181]
		*Kna1* locus, *TaALMT1* gene	4D	pentaploid hybridization, *ph1*-mediated homoeologous hybridization, homologous hybridization	bread wheat or Langdon-CS 4D (4B) substitution line	enhanced Al^3+^ tolerance	[182,183,184]
		/	/	intraspecific hybridization	*T. emmer*, *T. durum*, *T. carthlicum*, *T. turgidum*, *T. turanicum*, *T. polonicum*	enhanced salt tolerance	[185,186]
wheat-related disorders	coeliac disease	α-gliadin genes	/	CRISPR/Cas9 (knock out)	durum wheat	highest 69% reduction of gliadin of reduced immunoreactivity	[187]
	onset of bakers’ asthma	WTAI-CM3, WTAI-CM16	/	CRISPR/Cas9 (knock out)	durum wheat	reduced allergen proteins	[188]

/, unspecified; YPC, yellow pigment concentration; FHB, fusarium head blight; PM, powdery mildew; LR, leaf rust; SR, stem rust; YR, stripe rust; PHS, pre-harvest sprouting; LCYe, lycopene ε-cyclase; HYD2, β-carotene hydroxylase 2; HMW, high molecular weight; GW2, grain weight 2; WTAI, wheat α-amylase/trypsin inhibitors; CS, Chinese Spring.

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
