# Peer review of "Improvement and Re-Evolution of Tetraploid Wheat for Global Environmental Challenge and Diversity Consumption Demand"

_ijms, 2022, doi:10.3390/ijms23042206_

Round 1

Reviewer 1 Report

The review article entitled "Improvement and re-evolution of tetraploid wheat for global environmental challenge and diversity consumption demand" discussed the origin, domestication, and evolution of tetraploid wheat and investigated the breeding strategies and achievements using the gene introgression approach. The study is on relevance and general interest to the journal's readers. However, I have several concerns about presenting the data that should be addressed before publication.

  • The authors are highly recommended to avoid using a personal pronoun (e.g., We, our, etc.); they can use the third party in the past tense's passive voice.
  • The authors need to carefully read through the manuscript to correct typos and grammars to improve the manuscript.
  • Any abbreviation must be associated with the full name at the first mention in the manuscript to allow the reader to follow up because not all the readers are familiar with the abbreviated terminology.

Author Response

Point 1: The authors are highly recommended to avoid using a personal pronoun (e.g., We, our, etc.); they can use the third party in the past tense's passive voice.

 Response 1: Thank you for your valuable comments. We have changed them accordingly, such as lines 119 and 121.

Point 2: The authors need to carefully read through the manuscript to correct typos and grammars to improve the manuscript.

Response 2: Thank you for your reminder. We have thoroughly read and corrected these errors. All the changes were marked using the “Track Changes” function.

Point 3: Any abbreviation must be associated with the full name at the first mention in the manuscript to allow the reader to follow up because not all the readers are familiar with the abbreviated terminology.

Response 3: Thank you for your gentle reminder. We have revised them accordingly. For example, “Rht1” was changed to “reduced height gene Rht1 (Rht-B1b)” in line 662. All the changes were marked using the “Track Changes” function.

Reviewer 2 Report

There are some flaws in the manuscript that must be addressed. My main reservations about publishing the work in its current form are as follows:
Overall, the article contains less information every section has some scope of elobration.
To provide more specific information, the abstract should be revised.
Regardless of whether you use acronyms or not, please begin each phrase with the full form.
Despite the fact that the authors present a thorough evaluation of the literature in the Introduction, they should enrich and rebuild it in the following section. They must be clear about the scope of the paper they are writing.
A recurring theme is the repetition of information that was possibly overlooked the first time around.
Every topic should be thoroughly discussed. The vast majority of the sections are unfinished.
Enhance the summary section at the end.

Author Response

Point 1: Overall, the article contains less information every section has some scope of elobration.

 Response 1: We appreciate your valuable comments. Initially, our main focus was on the breeding achievements of durum wheat, therefore, we only give rather brief descriptions of the other sections. In the revised manuscript, we have followed your suggestion and added more information to these sections. All changes were marked using the “Track Changes” function.

Point 2: To provide more specific information, the abstract should be revised.

Response 2: We have revised the abstract (lines 16 – 32).

Point 3: Regardless of whether you use acronyms or not, please begin each phrase with the full form.

 Response 3: We have corrected them now throughout the manuscript. For example, “Rht1” was changed to “reduced height gene Rht1 (Rht-B1b)” in line 662. All changes were marked using the “Track Changes” function.

Point 4: Despite the fact that the authors present a thorough evaluation of the literature in the Introduction, they should enrich and rebuild it in the following section. They must be clear about the scope of the paper they are writing.

 Response 4: We have revised the manuscript accordingly by adding more information as well as restructuring the paragraphs to ensure that the points in the Introduction were enriched and rebuilt in the following sections. All changes were marked using the “Track Changes” function.

Point 5: A recurring theme is the repetition of information that was possibly overlooked the first time around.

Response 5: Thank you for your careful review. We have corrected them. For example, we deleted the description about the target characters of durum wheat improvement in the Introduction section (Line 144) but discussed it in the “3.2.2. Grain quality” section (Lines 904 - 905).

Point 6: Every topic should be thoroughly discussed. The vast majority of the sections are unfinished.

Response 6: Thank you for your valuable comments. During the revision, we have paid attention to this issue and have added some discussions to each topic. All changes were marked using the “Track Changes” function.

Point 7: Enhance the summary section at the end.

Response 7: Thank you for your suggestion. We have revised it. Please see lines 1600 – 1932.

Round 2

Reviewer 2 Report

Manuscript is now updated. Therefore it can be accepted for publication after a careful English language check.